# Assessing the Nutritional Value of Root and Tuber Crops from Bolivia and Peru

**DOI:** 10.3390/foods8110526

**Published:** 2019-10-23

**Authors:** Luz A. Choquechambi, Iber Roy Callisaya, Alvaro Ramos, Hugo Bosque, Angel Mújica, Sven-Erik Jacobsen, Marten Sørensen, Eduardo O. Leidi

**Affiliations:** 1Facultad de Ciencias Agrarias, Universidad Nacional del Altiplano, Ciudad Universitaria, Puno 51, Peru; solesolesol_lz@hotmail.com (L.A.C.); amhmujica@yahoo.com (A.M.); 2Facultad de Agronomía, Universidad Mayor de San Andrés, La Paz, Bolivia; roy-ce@hotmail.com (I.R.C.); hugobosque@yahoo.es (H.B.); 3Department of Plant Biotechnology, IRNAS-CSIC, E-41012 Seville, Spain; a.ramos@csic.es; 4Quinoa Quality, 4420 Regstrup, Denmark; quinoa@paradis.dk; 5Department of Plant & Environmental Sciences, University of Copenhagen, Thorvaldsensvej 40, 3, 1870 Frederiksberg C, Denmark; ms@plen.ku.dk

**Keywords:** arracacha, mashua, yacon, functional foods, fructoligosaccharides, glucosinolates

## Abstract

All over the world, there are species which may be considered as neglected or underutilized despite their nutritious properties. At present, such crops contribute to food security in isolated areas by providing energy and nutrients in a diversified diet. Such genetic heritage—improved by ancient cultures—is under threat of losing biodiversity as well as the traditional knowledge associated with their cultivation and usage. Among these species, the Andean root and tuber crops (ARTCs) constitute a valuable resource which should be preserved and popularized because of their food and functional properties. We studied three ARTC species (mashua, arracacha, and yacon) to provide data on their composition, essential for increasing their use globally. We compared their nutritional values with the values of more widely used crops. Important differences in nutrient composition among ARTC landraces were found. Mineral nutrients showed significant differences among species. Considerable variations in the contents of prebiotics like fructooligosaccharides or functional elements (antioxidants and glucosinolates) were found among species and intraspecific samples. Certainly, these species are important assets to complement human nutrition and to secure supply of functional elements for healthy diets.

## 1. Introduction

In the analysis of food security, food availability is one of the factors involved, but others like quantity, quality, and diversity of the food supply (in terms of energy and protein) are also important [1]. Only 30 crops out of the 30,000 edible plant species worldwide are used to provide 90% of the calories in the human diet [2]. Some of these species are used regularly for food by rural populations, but migration to urban centers has frequently led to the abandonment of traditional food sources. This dereliction occurs for several reasons, like difficulty in finding the products at local markets, cheaper sources of energy-rich basic foods (pasta, rice, and bread), the pressure of publicity, or by merely considering traditional foods as *poor man’s food* or *peasants’ food*, i.e., foods of low status [3]. The epidemic of overweight and obesity in Latin American children is partially a result of the dietary changes, with the intake of cheap processed food high in fat and sugars [4,5]. To reverse the situation, a change in the perception of low status of these minor crops is required, thereby increasing the awareness of their nutritional value [3]. Reinforcing (or sometimes reintroducing their uses) and avoiding abandonment of traditional recipes might contribute to the fight against malnutrition in the poorest city dwellers. At the same time, it will retain the rich crop heritage that has helped to maintain food security for centuries through agricultural systems rich in agrobiodiversity.

In the Andean region of South America, tuber-forming or storage root crops have been continuously domesticated from wild ancestors and improved via selection and breeding during centuries by ancient cultures [6,7]. Today, crops like mashua (*Tropaeolum tuberosum* Ruíz & Pav.), arracacha (*Arracacia xanthorrhiza* Bancr.), yacon (*Smallanthus sonchifolius* (Poepp.) H. Rob.), and other species are still important sources of carbohydrates, minerals, and vitamins for the local population in Bolivia, Ecuador, and Peru [8,9]. At present, however, they are all under the menace of losing both genetic diversity and the traditional knowledge associated with their crop husbandry and uses. These crops have significantly contributed to dietary diversification in rural communities by providing a balanced food supply containing all the elements of a healthy diet [10]. Simultaneously, they increase regional food security by promoting farm agrobiodiversity and diminishing the environmental risks of monoculture and genetic erosion [11]. Some international projects have dealt with the recovery of genetic diversity and traditional uses of these crops, e.g., Andescrop and Latincrop [12] in order to maintain the great richness in plant resources and local knowledge on cultivation and processing.

Root and tuber crops in the Andean region are complementary traditional food sources, but some have also been used for their healing properties [8,9,13]. This is of great interest when considering the growing awareness of diet diversification and the functional components and nutraceuticals in food [10,14]. The adoption of highly nutritious grains such as quinoa (*Chenopodium quinoa* Willd.) and amaranth (*Amaranthus caudatus* L.) by consumers of developed countries has boosted local economies in the Andean region [11]. Thus, it is reasonable to focus on the growing tendency to adopt other food crops, either as sources of functional elements [8] or exotic products for vegetarian and vegan consumers, in combination with a variety of new recipes [15]. Some of these crops are already marketed in countries like Japan and New Zealand for their attractive contents of functional components. The FAO INFOODS Database on underutilized crops provides some information (free and widely available) on the composition of these crops [16], but a more complete characterization of their nutritional components has yet to become available. Lately, some researchers have focused their attention on the evaluation of functional compounds in these species like glucosinolates, antioxidants, or fructooligosaccharides [17,18,19,20,21,22,23]. 

Our aim in the EU Project Latincrop was the compositional study of some root and tuber crops to provide quantitative data on their nutritional compounds to fill the missing gaps, e.g., FAO INFOODS Database [16]. We hope the results might help to scientifically back traditional uses for food and the claims of their healthy properties. We should bear in mind that rural populations still consume the crops presented in this report enriching rural diets with different functional compounds. Our principal objective is to increase the awareness of their nutritive qualities, either as vegetables or their derivatives, which might assist to promote their consumption worldwide.

## 2. Materials and Methods

### 2.1. Plant Materials

Three different ARTC species, namely, mashua (*T. tuberosum*), arracacha (*A. xanthorrhiza*), and yacon (*S. sonchifolius*), were collected in Peru and Bolivia (see Appendix A) from farmers’ fields and composed of 10–15 subsamples. Two additional harvests were performed during 2016 and 2017 in Bolivia to correct initial errors in sample preparation for mineral analyses. Two different mashua landraces (*chiar* and *kellu*) were cultivated in Seville (Spain) for performing analytical controls on glucosinolate contents. All samples were washed, peeled, freeze-dried, and kept at −20 °C until analyses. In samples from Peru, tissue pieces of the three species were blanched with hot water for 2 min.

### 2.2. Minerals

The concentration of N in the samples was determined after Kjeldahl digestion in a Technicon autoanalyzer. The remaining macro and micronutrients were analyzed after acid digestion with nitric acid in a microwave digestor (Novawave, SCP Science) by inductively coupled plasma-optical emission spectrometry (Varian ICP 720-ES).

### 2.3. Starch Analyses

Starch was measured following sample dilution and hydrolysis recommended in R-Biopharm’s kit, Starch (Boehringer Mannheim, Darmstadt, Germany). Ground samples were dissolved with dimethylsulfoxide and 8 M hydrochloric acid by incubation at 60 °C for 1 h, cooled quickly, and adjusted to pH 4–5. Starch hydrolysis was performed with amyloglucosidase to produce d-glucose. d-glucose was then determined by NADPH formation after incubation with hexokinase and glucose-6-phosphate dehydrogenase.

### 2.4. Fructooligosaccharides

Water-soluble carbohydrates were extracted from freeze-dried samples with hot Milli-Q water (90 °C, 1 h), centrifuged (13,000× *g*, 10 min), and filtered (0.45 µm). Fructan analysis was performed by the acetone precipitation method [24]. An aliquot of the filtrate was treated with acetone (8 vol acetone:1 vol filtrate) and left overnight at 4 °C to precipitate fructan molecules. The precipitate was concentrated by centrifugation (25 min, 13,000× *g*) and hydrolyzed by adding HCl to a final concentration of 150 mM and incubation at 80 °C for 90 min. Then, glucose, fructose, and sucrose were measured using the enzymatic kits described below.

### 2.5. Sugars

Sugars were measured in samples after extraction with hot water (90 °C, 1 h). Enzymatic kits from R-Biopharm for sucrose, d-glucose, and d-fructose were used.

### 2.6. Organic Acids

The root and tuber contents of oxalate were analyzed after extracting 0.5 g sample with 10 mL distilled H_2_O at pH 3.0 (by adding 1.5 M HCl) in a water-bath at 100 °C during 15 min. The analysis of oxalate was performed after passing supernatants through active carbon columns for color and antioxidant removal (Enzytec oxalic acid kit, R-Biopharm). Malate and citrate were extracted from lyophilized samples in water and determined using enzymatic kits (l-malic and citric acid from R-Biopharm).

### 2.7. Lipids and Fibres

Lipids were extracted with *n*-hexane and gravimetrically measured according to ISO 659: 2009. Fibres were analyzed using heat-stable α-amylase, protease, and amyloglucosidase [25].

### 2.8. Protein Hydrolysis and Amino Acid Analysis

Freeze-dried samples were dissolved in 6.0 M HCl with d,l-α aminobutyric acid as internal standard. The samples in HCl acid were gassed with nitrogen, sealed in hydrolysis tubes with nitrogen, and then incubated in an oven at 110 °C for 24 h. Derivatization and chromatography of amino acids was performed [26]. Dried samples of protein hydrolysates were dissolved in 1 M sodium borate buffer (pH 9) and derivatized with diethyl ethoxymethylenemalonate. Separation was performed in a reversed-phase column using sodium acetate and acetonitrile as eluents [26,27]. Tryptophan was measured separately by HPLC after alkaline hydrolysis of samples [28].

### 2.9. Protein Determination

Protein contents in the samples were estimated as the concentration of amino acids after protein hydrolysis (in g amino acids 100 g dried sample^−1^) minus the concentration of free amino acids.

### 2.10. Other Functional Compounds: Anthocyanins, Carotenoids, Phenolic, and Glucosinolates

Total anthocyanins, carotenoids, and phenolic compounds were similarly determined as reported in [18,19]. In short, anthocyanins were homogenized in 85 parts ethanol:15 parts 1.5 M HCl, left overnight at 4 °C, extracting carotenoids with *n*-hexane and determining anthocyanins by spectrophotometry (A_535nm_). Total carotenoids were extracted with acetone:ethanol (1:1), left in darkness overnight, extracted with n-hexane, and measured at A_470nm_. Total phenols were extracted from lyophilized samples in 96% ethanol and determined using diluted Folin-Ciocalteu reagent and 0.5 M sodium carbonate and measured at A_750nm_. A standard curve with chlorogenic acid was used to estimate total phenols in the samples. Glucosinolate content in mashuas was determined by HPLC according to AOCS Official Method [29].

### 2.11. Statistical Analyses

Collected data were analyzed using a commercial statistical package (Statistica) in a completely randomized design. After ANOVA, means comparison was performed when significant *F* was recorded following least significant difference (LSD) test at *p* < 0.05. Several difficulties arising from national protection laws on local biodiversity limited the collection of materials and their transfer out of the countries of origin for laboratory analysis. Different varieties of the same species were treated as replicates because of the low number of independent samples. Each determination was performed two to three times (technical replicates).

## 3. Results and Discussion

The different ARTC species averaged across countries and varieties, and showed significant differences in the concentration of essential mineral nutrients (Table 1). The mashuas contained a higher concentration of N, Mg, P, and S and the essential micronutrients Fe, Zn, and Mn than the remaining species (Table 1). Meanwhile, yacon roots showed the highest content in Ca, B, Ni, and Cu. The content of Na was low in all the three crops, while the K content was somewhat similar between them (Table 1). Interestingly, arracacha—previously described as having high P, Fe, and Ca content [30,31]—showed lower content of P and Fe than mashua and Ca than yacon (Table 1). Different treatments before chemical analysis, such as root or tuber peeling in previous reports [30,31] which may significantly affect the concentration of nutrients like K and Fe, might explain these differences. In comparison with other roots and tubers (potato (*Solanum tuberosum* L.), manioc/cassava (*Manihot esculenta* Crantz), and greater yam (*Dioscorea alata* L.) [32,33], the ARTC species present greater concentration in essential mineral nutrients, mostly in K and P and the micronutrients Fe and Zn.

The lipid and insoluble fiber content in all the three species (Table 2) were rather high in comparison with potato [32], but comparable to other root and tuber crops [33]. The highest content of fibers was found in mashua, while arracacha showed the lowest content of lipids and yacon presented the lowest fiber content out of all the three species (Table 2).

Significant differences among ARTC species were also found for starch content (Table 3), with arracacha showing the highest starch content, while the lowest values were recorded in yacon roots. The high starch yield in arracacha and its particular properties (highest amylopectin content among other ARTCs, easily cooked, and high digestibility) [30,34,35] are the reason for its uses in food recipes for infants and elderly people [31] or more recent applications in the food industry [36]. However, the use of arracacha for food in the European Union has been rejected by European Food Safety Authority as it might contain potentially allergenic compounds like terpenes or coumarin [37]. Arracacha and mashua, mostly used in stews [9,31], are low-fat sources of energy but provide other functional components such as carotenoids as well, however, with significant variation between cultivars [18,31]. As it might be expected, fructooligosaccharides (FOS) were only found in yacon samples though with significant differences among landraces (Table 3). The significance of dietary carbohydrates like FOS, which are not hydrolyzed in the small intestine, is through their prebiotic role in stimulating the growth of beneficial bacteria in the colon [14,38]. The high sugar concentration (mostly fructose) in yacon from Bolivia (Table 3) might be due to the traditional sun exposure of roots for increasing their sweetness [30,39,40]. The rate of FOS breakdown—by hydrolysis into fructose after sunlight exposure and the final concentration of sugars—is also dependent on the landrace [40]. This fact should be taken into account when its use is intended for diabetics or people suffering from fructose intolerance or other metabolic disorders [41] or when roots are to be used as a source of industrial prebiotics. The concentration of sucrose and glucose was somewhat variable between species and landraces of the same species, but the lowest concentration of monosaccharides and disaccharides was found in arracacha (Table 3).

In the ARTC samples studied, significant variation was found in the concentration of two organic acids, malate and citrate (Table 4), while the concentration of oxalate was low and similar. The concentration of oxalate, a food component that should be taken into account for its role in urolithiasis (kidney stone disease), [42] was lower than that found in potato or sweet potato (*Ipomoea batatas* (L.) Lam.) [43]. The amount of ascorbate was low and negligible in arracacha (Table 4). The stability of ascorbate, i.e., vitamin C, was maintained after freeze drying of samples, although further storage temperature and time decreased ascorbate content [44]. It might be the reason for the generally low content of vitamin C in the samples.

The carotenoid content was higher in mashuas, with a significant intraspecific variation (Table 5). Arracacha root samples from Bolivia and Peru presented a somewhat similar concentration in carotenoids (Table 5) whereas, in yacon roots, carotenoids were mostly found in the orange-colored landrace, *kulli* Mocomoco from Bolivia (Table 5). Anthocyanins were almost absent in arracachas, while a significant variation was observed for anthocyanin content in mashua and yacon samples (Table 5). It seemed that anthocyanin was the main compound providing color in the darkest colored mashua landrace (*chiar* or black mashua). Total phenolic compounds also showed interspecific variation in concentration (Table 5), with high content in mashuas and yacon landraces and low content in arracacha. Previous reports on antioxidant activity in mashua tubers [19,20,21] found significant variation among genotypes in phenolic compounds, anthocyanins, and carotenoids, but also a variation depending on time to harvest and post-harvest storage conditions.

Glucosinolate analyses in mashua showed a significant variation in content among samples, from very low values, almost in the limit of detection, to rather high values, which occurred in freshly harvested tubers (Table 6). The only peak identified in the different mashua landraces was desulfoglucoaubrietin as reported [17]. The considerable variation in glucosinolate content among samples (Table 6) might indeed correspond to genotypic variation already reported in cultivated and wild mashuas [45]. However, the significant variation in concentration observed in Bolivian samples from year to year might be due to compound degradation during sample preparation or improper storage [17]. Mashua glucosinolate is the main compound with biological activity related to traditional uses for controlling prostate cancer, libido inhibitors, or female fertility [46]. More recent pharmacological and epidemiological studies associate diet glucosinolates with cancer prevention [47,48,49]. Among several effects, some glucosinolate derivatives alter microRNA expression, which in turn suppresses prostate cancer cell invasiveness [50].

The protein content, determined as the total of amino acids after acid hydrolysis, was the highest in mashua (Table 7), while arracacha and yacon presented similar protein values. In other root and tuber crops (potato, manioc/cassava, sweet potato, and greater yam), proximate protein analysis (Kjeldahl N concentration × 6.25) provided similar data to arracacha and yacon [33]. However, using the same calculation method, a significant variation in protein concentration was reported for our three species as well for other ARTCs [30]. Previous results with other tuber crops have pointed out the high concentration of soluble N as a factor for providing an inaccurate estimate of protein content [51,52]. Probably, there is a significant variation due to the genetic diversity of landraces as has been reported previously [53]. The most exciting feature of mashua proteins relies on their higher concentration in sulfur-containing essential amino acids (methionine and cysteine) than arracacha and yacon, as well in other essential amino acids like lysine, histidine, tyrosine, valine, leucine, isoleucine, and phenylalanine (Table 7).

Interestingly, free amino acids might also provide essential amino acids and functional amino acids (arginine and aspartate) [54]. The contents of free amino acids showed greater interspecific and intraspecific variations, resulting in statistically significant differences among crops only in the content of glutamine (yacon), arginine and methionine (arracacha), and valine (mashua) (Table 8). An approach to the protein quality of each crop based on amino acid scoring of their proteins (Table 9) showed that mashua might provide a complete source of essential amino acids, while arracacha and yacon are somewhat deficient in cysteine and methionine, and yacon is more deficient in lysine and tryptophan (Table 9). Root and tuber crops, mostly mashua, might supply essential amino acids in the diet even though they do not hold a high protein content. Diets rich in carbohydrates (e.g., wheat bread and pasta) may require supplementation of essential amino acids like lysine and threonine [55]. In this sense, tuber crops like mashua may be significant sources of many essential amino acids which might supplement amino acid deficient-diets of people relying mostly on bread and pasta as energy sources.

All the three species show a significant variation in basic food components in the form of energy sources (starch and sugars) or functional elements (essential amino acids, vitamins, FOS, glucosinolates, and antioxidants). At present, they constitute an important contribution to balance the diet of local farmers in the Andes, while increasing their food security standards. However, they could potentially also provide a significant contribution to healthy diets for urban populations in the Andean region, i.e., populations already affected by increasing rates of obesity and non-communicable diseases [5]. They may provide alternative sources of novel products for vegetarian, vegan, and novel cuisine approaches [15]. Yacon, as of now considered a novel food after fulfilling safety standards, is already being produced and marketed for controlling bodyweight or dietary supplement in hyperglycemic patients [57,58,59]. The use of yacon syrups or inclusion of yacon flour in fermented milks has beneficial effects when adopted in healthy diets by improving organoleptic properties of dairy products [59,60,61] and the intestinal absorption of Ca and Fe [62]. Even though other compounds, like mashua glucosinolates, might provide nutraceutical properties on prostate affections attributed by folk medicine [63], they should await further studies involving humans. The same opinion should be held for mashua or yacon antioxidants [20,21,23,64] and their relationship with human health [65]. Arracacha is another crop with a long history of safe use all over Latin America, although it awaits analyses in potentially allergenic compounds.

At present, food and functional properties of the three species have been mostly supported by traditional uses or research using different experimental approaches [9]. Meanwhile, as part of traditional or new recipes [15,66], tubers like mashua provide essential mineral nutrients and other functional elements like amino acids which complement diets low in animal proteins or protein-rich plant seeds (e.g., legumes and quinoa).

The shift from minor to cash crops—mostly soybean (*Glycine max* (L.) Merr.), maize (*Zea mays* L.), or wheat (*Triticum aestivum* L.), or where coca (*Erythroxylum coca* Lam.) replaces arracacha in some Bolivian regions—provides money to peasants but compromises their nutritional security and agrobiodiversity [67]. The promotion of consumption of all the three Andean species would reinforce the maintenance of such valuable crops menaced by diversity loss, at a time when it could increase the economic income of rural communities as occurred successfully with quinoa [68]. From a merely nutritional perspective, they enrich the diet diversity of local communities, with their health-associated benefits, and supply additional food components for balanced nutrition. Moreover, last but not the least, their significant value for food security in the region as hardy crops very well adapted to climatic variation and mountain agriculture merits their advancement [6]. These minor crops are produced under low-cost traditional farming conditions and have been securing basic and nutritious food supply for centuries. Indeed, based on the ARTC content in functional components, they are also attractive food sources in a worldwide market now open to discover new healthy foods. Functional foods with qualified health claims are required in a world where obesity and other diet-related diseases are now of major public concern [69]. At present, they provide nutritious and safe foods which support food security in the Andean region. By being sustainable crops per se as part of traditional agricultural systems that maintain high agrobiodiversity [11], increasing their consumption might contribute to boost local economies at a time when sustainability of both food and environment are matters of increasing global concern [70,71].

## Figures and Tables

**Table 1 foods-08-00526-t001:** Mineral nutrient composition of Andean roots and tubers collected in Bolivia and Peru in 2015, 2016, and 2017. For each species, averages across countries and varieties. Among species, results with different letters are significantly different at *p* < 0.05 (LSD).

Crops	N	K	Ca	Mg	P	S	Na
	g 100 g dry matter^−1^						
Mashua	1.41 ^a^	1.47 ^a^	0.05 ^a^	0.13 ^a^	0.56 ^a^	0.39 ^a^	0.008 ^a^
Arracacha	0.46 ^b^	1.41 ^a^	0.06 ^a^	0.05 ^b^	0.21 ^b^	0.09 ^b^	0.002 ^a^
Yacon	0.35 ^b^	1.41 ^a^	0.14 ^b^	0.07 ^b^	0.22 ^b^	0.11 ^b^	0.060 ^a^
**Crops**	**Fe**	**Zn**	**Mn**	**Cu**	**Ni**	**Co**	**B**
	**mg kg dry matter** ^**−1**^						
Mashua	48.3 ^a^	29.5 ^a^	12.6 ^a^	4.7 ^a^	1.19 ^a^	0.22 ^a^	07.6 ^a^
Arracacha	14.7 ^b^	07.3 ^b^	03.4 ^b^	2.3 ^b^	0.41 ^b^	0.10 ^a^	06.5 ^a^
Yacon	19.8 ^b^	10.9 ^b^	09.3 ^c^	5.5 ^a^	1.36 ^a^	0.08 ^a^	10.1 ^b^

**Table 2 foods-08-00526-t002:** Variation in lipid and fiber contents in Andean roots and tubers g 100 g DM^−1^). For each species, averages across countries and varieties (g 100 g dry matter^−1^ or mg kg dry matter^−1^). Among species, results with different letters are significantly different at *p* < 0.05 (LSD).

	Mashua	Arracacha	Yacon
Lipids	0.63 ± 0.06 ^a^	0.43 ± 0.08 ^b^	0.50 ± 0.08 ^ab^
Fibers	6.9 ± 0.1 ^a^	4.7 ± 0.1 ^b^	4.1 ± 0.1 ^c^

**Table 3 foods-08-00526-t003:** Starch, fructooligosaccharides (FOS), and sugars in samples of mashua, arracacha, and yacon from Bolivia and Peru (g 100 g dry matter^−1^).

	Starch	FOS	Sucrose	Glucose	Fructose
**Peru**					
Mashua					
*Kellu*	45.5	--	37.4	09.2	--
*Chejchi*	39.5	--	36.2	10.3	--
Arracacha					
Purple heart	67.0	--	15.6	03.0	--
Yacon					
White	18.3	43.3	35.7	14.5	04.7
**Bolivia**					
Mashua					
*Chiar*	32.4	--	21.6	9.0	--
*Kellu*	17.4	--	45.4	14.9	--
*Keni kellu*	30.7	--	32.1	15.1	--
*Jachir*	34.0	--	35.3	16.3	--
*Asuthi*	22.2	--	43.9	20.1	--
Yacon					
*Kulli Mocomoco*	14.2	42.7	41.2	18.7	20.7
*Kulli Sorata*	19.7	17.3	31.9	10.4	27.8
Blanco	12.3	37.0	26.0	08.1	31.8
Arracacha					
San Juan Miel	49.3	--	07.2	00.9	--

-- Not detected.

**Table 4 foods-08-00526-t004:** Concentration of organic acids in Andean roots and tubers collected in Bolivia and Peru (in mg g dry matter^−1^). Means followed with the same letter indicate not significant differences among species (LSD, *p* < 0.05).

	Mashua	Arracacha	Yacon
Ascorbate	0.16 ± 0.1 ^a^	0.004 ± 0.02 ^a^	0.27 ± 0.1 ^a^
Oxalate	1.5 ± 0.7 ^a^	1.1 ± 1.8 ^a^	2.7 ± 1.2 ^a^
Malate	5.6 ± 0.5 ^a^	5.8 ± 1.0 ^a^	2.8 ± 0.8 ^b^
Citrate	4.5 ± 0.7 ^a^	9.2 ± 1.8 ^b^	1.7 ± 1.3 ^a^

**Table 5 foods-08-00526-t005:** Concentration of carotenoids, anthocyanins, phenols, and glucosinolates in Andean root and tuber samples.

	Carotenoids	Anthocyanins	Phenols
	(µg β Carotene g Dry Matter^−1^)	(mg g Dry Matter^−1^)	(mg Chlorogenic Acid g Dry Matter^−1^)
**Peru**			
Mashua			
*Kellu*	28.6	0.01	07.9
*Chejchi*	13.9	1.03	11.9
Arracacha			
Purple yellow	07.2	0.02	01.6
Yacon			
White	00.0	0.01	15.5
**Bolivia**			
Mashua			
*Chiar*	21.9	3.63	22.3
*Kellu*	50.1	0.08	12.8
*Keni kellu*	96.7	0.11	13.1
*Jachir*	66.1	0.05	09.7
*Asuthi*	61.5	0.37	08.0
Arracacha			
Yellow	09.6	00.0	03.7
Yacon			
*Kulli Mocomoco*	19.3	0.49	14.3
*Kulli Sorata*	03.0	0.34	17.8
White	02.6	0.03	05.9

**Table 6 foods-08-00526-t006:** Variations in glucosinolate (desulfoglucoaubrietin) content in mashua tubers collected in Peru and Bolivia from 2015 to 2017. Included, control values from tubers harvested in Spain.

	Glucosinolates (µmol g Dry Matter^−1^)
**Peru 2015**	
*Kellu*	29.7
*Chejchi*	46.5
**Bolivia 2015**	
*Chiar*	04.4
*Kellu*	18.3
*Keni kellu*	14.5
*Jachir*	63.5
*Asuthi*	35.7
**Bolivia 2016**	
*Chiar*	0.10
*Kellu*	0.23
*Keni kellu*	0.03
**Bolivia 2017**	
*Chiar*	0.11
*Kellu*	0.06
*Jachir*	0.12
*Asuthi*	0.05
**Spain 2018**	
*Chiar*	83.0
*Keni kellu*	96.6

**Table 7 foods-08-00526-t007:** Protein and protein amino acid concentration in mashua, arracacha, and yacon from Bolivia and Peru (in g 100 g sample^−1^). Means of different cultivars (mashua, 7; arracacha, 2; yacon, 2) ± standard deviation.

	Mashua	Arracacha	Yacon	Significance
Proteins	5.14 ± 1.05	1.97 ± 0.49	1.89 ± 0.06	*p* < 0.01
Amino acids				
Aspartic + Asparagine	2.47 ± 1.48	0.60 ± 0.07	0.31 ± 0.01	ns
Glutamic + Glutamine	0.68 ± 0.08	0.39 ± 0.04	0.72 ± 0.03	*p* < 0.01
Serine	0.36 ± 0.08	0.13 ± 0.04	0.10 ± 0.01	*p* < 0.01
Histidine	0.20 ± 0.05	0.08 ± 0.02	0.05 ± 0.001	*p* < 0.01
Glycine	0.35 ± 0.09	0.10 ± 0.03	0.09 ± 0.001	*p* < 0.01
Threonine	0.33 ± 0.06	0.11 ± 0.04	0.09 ± 0.001	*p* < 0.001
Arginine	0.41 ± 0.14	0.81 ± 0.14	0.62 ± 0.02	*p* < 0.05
Alanine	0.47 ± 0.08	0.27 ± 0.13	0.11 ± 0.01	*p* < 0.01
Proline	0.49 ± 0.17	0.27 ± 0.02	0.41 ± 0.01	ns
Tyrosine	0.21 ± 0.05	0.05 ± 0.001	0.03 ± 0.001	*p* < 0.01
Valine	0.78 ± 0.26	0.05 ± 0.05	0.09 ± 0.01	*p* < 0.01
Methionine	0.05 ± 0.02	0.005 ± 0.001	0.00	*p* < 0.01
Cysteine	0.04 ± 0.01	0.005 ± 0.01	0.01 ± 0.001	*p* < 0.01
Isoleucine	0.40 ± 0.09	0.09 ± 0.04	0.08 ± 0.004	*p* < 0.001
Tryptophan	0.05 ± 0.02	0.09 ± 0.12	0.01 ± 0.001	ns
Leucine	0.46 ± 0.09	0.13 ± 0.05	0.11 ± 0.01	*p* < 0.001
Phenylalanine	0.36 ± 0.07	0.12 ± 0.05	0.07 ± 0.001	*p* < 0.001
Lysine	0.45 ± 0.09	0.13 ± 0.04	0.05 ± 0.06	*p* < 0.001

ns, not significant.

**Table 8 foods-08-00526-t008:** Free amino acids in mashua, arracacha, and yacon from Bolivia and Peru (in g 100 g sample^−1^) and statistical significance of differences among species. Means of different cultivars (mashua, 7; arracacha, 2; yacon, 2) ± standard deviation.

	Mashua	Arracacha	Yacon	Significance
Aspartic acid	0.07 ± 0.04	0.09 ± 0.02	0.05 ± 0.001	ns
Glutamic acid	0.09 ± 0.06	0.06 ± 0.06	0.08 ± 0.001	ns
Asparagine	1.51 ± 1.67	0.22 ± 0.001	0.11 ± 0.002	ns
Glutamine	0.04 ± 0.04	0.10 ± 0.02	0.44 ± 0.001	*p* < 0.001
Serine	0.07 ± 0.07	0.04 ± 0.005	0.02 ± 0.001	ns
Histidine	0.05 ± 0.04	0.03 ± 0.01	0.01 ± 0.001	ns
Glycine	0.01 ± 0.003	0.003 ± 0.001	0.002 ± 0.000	*p* < 0.01
Threonine	0.06 ± 0.03	0.03 ± 0.02	0.02 ± 0.000	ns
Arginine	0.38 ± 0.15	0.58 ± 0.19	0.15 ± 0.001	*p* < 0.05
Alanine	0.14 ± 0.08	0.12 ± 0.07	0.02 ± 0.001	ns
Proline	0.16 ± 0.16	0.02 ± 0.001	0.26 ± 0.002	ns
Tyrosine	0.07 ± 0.05	0.01 ± 0.001	0.001 ± 0.000	ns
Valine	0.33 ± 0.19	0.01 ± 0.003	0.001 ± 0.001	*p* < 0.05
Methionine	0.00	0.006 ± 0.001	0.00	*p* < 0.001
Cysteine	0.00	0.00	0.00	--
Isoleucine	0.12 ± 0.10	0.01 ± 0.003	0.01 ± 0.001	ns
Tryptophan	0.04 ± 0.02	0.02 ± 0.02	0.01 ± 0.001	ns
Leucine	0.07 ± 0.08	0.02 ± 0.03	0.002 ± 0.001	ns
Phenylalanine	0.09 ± 0.08	0.03 ± 0.02	0.003 ± 0.000	ns
Lysine	0.06 ± 0.05	0.02 ± 0.003	0.003 ± 0.000	ns

ns, not significant.

**Table 9 foods-08-00526-t009:** Quality assessment for mashua, arracacha, and yacon proteins based on their essential amino acid scoring pattern.

	Amino Acid Pattern ^1^	Mashua	Arracacha	Yacon
His	27	144	109	98
Ile	35	222	130	121
Leu	75	119	88	78
Lys	73	120	90	36
SAA ^2^	35	50	14	15
AAA ^3^	73	118	118	72
Thr	42	153	133	113
Trp	12	81	109	13
Val	49	309	52	97

^1^ Tissue amino acid pattern based on amino acid composition of whole-body protein (in mg g protein^−1^). Source: [56]. ^2^ SAA, sulfur amino acids (met + cys); ^3^ AAA, aromatic amino acids (phe + tyr).

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
