# Peer review of "Assessing the Nutritional Value of Root and Tuber Crops from Bolivia and Peru"

_foods, 2019, doi:10.3390/foods8110526_

Round 1

Reviewer 1 Report

The manuscript entitled “Assessing the nutritional value of root and tubers crops from the Andes” submitted for revision in journal Food needs correction.

The title should be changed because test samples were collected from Peru and Bolivia. I suggest:

Assessing the nutritional value of root and tubers crops from Peru and Bolivia.

The material and methods chapter should be revised. Plant materials: there is no information on the detailed location of the samples, from which region/place they were taken. How many samples were collected for analysis, in what period? Minerals: Invalid description - there is no description of the method for determining mineral components (K, Ca, Mg, P, S, Na Fe, Zn, Mn, Cu, Ni, Co, B). What method and on which analyzers were determined? The description in line 99 refers to the determination of protein and not minerals. Protein/Kjeldahl N concentration should be given and described in the part regarding materials and methods (there was a calculator × 6.25?) Statistical analyses: The description needs to be completed. Which kind of statistical tests were used to analyze which level of significance was adopted?

In many cases, probably due to the small number of attempts to analyze, the results are strange.

For example: In Table 4, the statistical description is primarily incorrect because there is an incredible difference between the mean value. The same applies to the data in Table 7 (Aspartic + Asparagine) Table 8 (Asparagine, Isoleucine, Leucine, Phenylalanine, Lysine). It is likely that these results are determined by a small number of samples. The authors should explain it.

Standard deviation should be added to tables 7 and 8.

Author Response

The title has been changed as suggested because he/she is right: the Andean region is much wider and several countries are in it. The detailed information on samples collection appeared in a Supplementary Table provided in pdf format. Information on harvest dates reported by the collectors is also included now. The mineralization for ICP analysis is now included in M&M, The method used for mineralization with nitric acid correspond to the protocol provided by the microwave digestor. Protein content in the samples was computed by considering the amino acid concentration  after protein digestion minus free amino acids concentration as stated in M&M. Details on statistical treatment of data are now reported. The reviewer was right, and the only mention on significance level was provided in some Tables. I do not understand completely the objection on results presented in Table 4. It might be that the number of samples were not enough, but other sources of variation like time of sample collection and storage conditions (mainly temperature or sun exposure) before freezing/lyophilization. It was our main interest to receive a high number of accessions and reps but existing border controls and available funds for analyses have not permitted a wider sampling. Standard deviation has been added in Tables 7 & 8.

Reviewer 2 Report

The manuscript provides data related with the nutritional and mineral composition and functional elements of three Andean roots and tuber crops (mashua, arracacha and yacon).

From the point of view of innovation, the characterization of the roots and tubers corresponds to the application of methodologies that do not add an innovative character to the work. However, the data obtained in this work are important to complement the results published in the literature about these neglected and underutilised crops and to contribute to increasing the awareness of the nutritional and functional qualities of the crops under study. The better knowledge of these crops can scientifically demonstrate the traditional uses for food and establish health claims that promote their consumption and valorisation not only in Andean region but also around the worldwide market.

The paper is generally well written.  I suggest only the following minor revision points:

Explicit the meaning of DM in the units of chemical composition of crops; Table 5: increase the width of the columns; Table 9: put “Amino acid pattern” only in a row.

Author Response

The meaning of DM is now included in each Table as 'dry matter' in the corresponding, and corrected in other Tables were erroneously appeared in Spanish (MS). The heading in Table 9 was corrected and now appears in the same row.

Round 2

Reviewer 1 Report

The authors have improved as suggested by the reviewer.